# TENSOR-TRAIN POINT CLOUD COMPRESSION AND EFFICIENT APPROXIMATE NEAREST NEIGHBOR SEARCH

## ABSTRACT

Nearest-neighbor search in large vector databases is crucial for various machine learning applications. This paper introduces a novel method using **tensor-train** (TT) low-rank tensor decomposition to efficiently represent point clouds and enable fast approximate nearest-neighbor searches. We propose a probabilistic interpretation and utilize density estimation losses like Sliced Wasserstein to train TT decompositions, resulting in robust point cloud compression. We reveals an inherent hierarchical structure within TT point clouds, facilitating efficient approximate nearest-neighbor searches. In our paper, we provide detailed insights into the methodology and conduct comprehensive comparisons with existing methods. We demonstrate its effectiveness in various scenarios, including out-of-distribution (OOD) problems and approximate nearest-neighbor (ANN) search tasks.

## 1 INTRODUCTION

Nearest-neighbour search in large vector databases (clouds of high-dimensional points) constitutes a fundamental component in numerous computer science applications, like local or global image matching, semantic search, out-of-distribution and anomaly detection, among others.

In this paper, we propose to use *tensor-train* (TT) Oseledets (2011) low-rank tensor decomposition to represent point clouds and to facilitate fast approximate nearest-neighbour search. The TT decomposition method has gained prominence in deep learning, primarily to compress large tensors of neural network parameters. When representing point clouds as matrices, we artificially order vectors and thus direct application of TT to compress this matrix with standard tensor approximation algorithms (such as TT-SVD Oseledets (2011); Cichocki et al. (2016), TT-cross approximation Oseledets & Tyrtyshnikov (2010) or ALS Holtz et al. (2012)) would result in a very poor quality. To address this issue, we propose probabilistic interpretation of the point cloud. Specifically, we suggest training TT to compress point cloud using loss functions from density estimation, notably Sliced Wateriness loss, through standard gradient descent method: Section 2.

Moreover, we observe the ability of the TT point cloud to implicitly contain inner hierarchical structure of points: we can consider it as a hierarchical KMeans clusterization and TT format allows fast computation of centroids of different levels. This observation allows us to build a highly efficient approximate nearest-neighbors search algorithm on top of the compressed TT point cloud: Section 2.

Due to the nature of the proposed method, points inside compressed TT point cloud do not have a direct correspondence with points in original point cloud. However, in out-of-distribution (OOD) detection methods that use distance to the normal point cloud as a score, this is not a problem. The state-of-the-art method that builds upon this methodology is PatchCore Roth et al. (2022). Patchcore utilizes greedy coreset subsampling to reduce redundancy in the point cloud and required storage memory. We demonstrate how to use TT point cloud compression in this setting in Section 2 and show that our method can significantly outperform coreset-based point cloud subsampling in Sections 2 and 4.2.

For the case of nearest-neighbour search, where the output of the search should be from the original point cloud we suggest to use TT as a search index structure as a replacement for indexes like FAISS Johnson et al. (2017), IMI Babenko & Lempitsky (2015), (G)NO-IMI Babenko & Lempitsky (2016), as detailed in Sections 2 and 4.3.

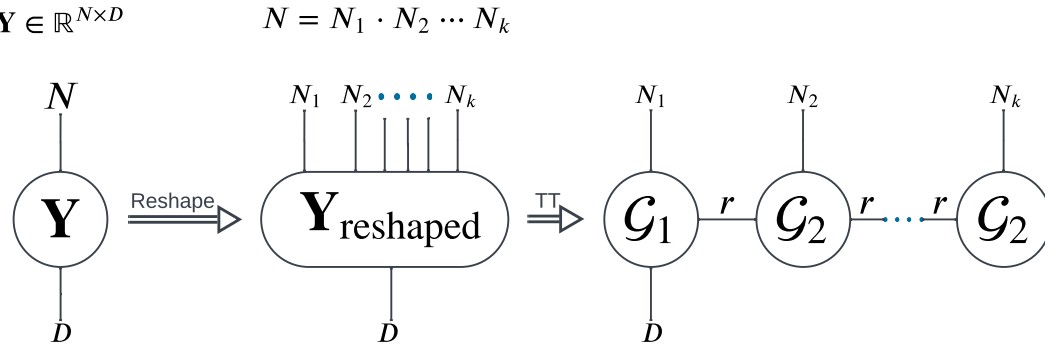

Figure 1: Diagram of the TT point cloud in Penrose graphical notation. Each tensor is depicted as a vertex, and each vertex has as many edges as the dimensionality of the corresponding tensor. Two tensors are connected with a common edge if these two tensors are contracted along the corresponding dimension.

## 2 METHOD

**Point Cloud Tensorization.** In the context of the OOD detection based on normal (in-distribution) points or ANN problems requiring k-nearest-neighbor search, we begin by considering a set of $D$-dimensional vectors $\mathcal{Y} = \left\{ \boldsymbol{y}_i \in \mathbb{R}^D \right\}_{i=1}^{N}$. It is a natural approach to store this set as a matrix $\boldsymbol{Y} \in \mathbb{R}^{N \times D}$, such that $\boldsymbol{y}_i$ corresponds to the $i$-th row of matrix $\boldsymbol{Y}$.

Low-rank tensor methods are wide popular in tensor compression and representation. We chose *tensor-train* (TT) decomposition due to its efficiency and computational synergies with our method. TT decomposition of a matrix $\boldsymbol{Y}$ starts with matrix tensorization, which involves reshaping matrix into a high-dimensional tensor. In our specific case, we factorize rows dimension (samples dimension) of matrix $\boldsymbol{Y}$: $N = N_1 \times N_2 \times \cdots \times N_k$ while leaving the columns dimension (features dimension) $D$ unchanged:

$$\mathcal{Y}_{\text{reshaped}} = \text{reshape}(\boldsymbol{Y}, [N_1, \cdots, N_k, D]) \tag{1}$$

Low-rank decomposition of tensor $\mathcal{Y}_{\text{reshaped}}$ is parameterized by $k$ 3-dimensional tensors, referred to as tensor-train cores. The first core, $\mathcal{G}_1 \in \mathbb{R}^{D \times N_1 \times r_1}$, factorizes out feature dimension $D$ and the first sample dimension $N_1$. The remaining $k-1$ cores, $\mathcal{G}_i \in \mathbb{R}^{r_{i-1} \times N_i \times r_i}$, separate all remaining sample dimensions $N_2, \cdots, N_k$:

$$\mathcal{Y}_{\text{reshaped}}[i_1, \cdots, i_k, d] = \mathcal{G}_1[d, i_1, :]\mathcal{G}_2[:, i_2, :] \cdots \mathcal{G}_k[:, i_k, :]. \tag{2}$$

Here, $(i_1, \cdots, i_k)$ represents a multiindex of a sample, which replaces the standard index $i$ after reshape. The dimensions $r_1, \cdots, r_k$ are known as *tensor-train ranks* (TT-ranks) and are subject to the constraint that $r_k = 1$. While TT-ranks are hyperparameters and in general can be different, in our work, we take $r_1 = \cdots = r_{k-1} = r$. The entire TT construction is depicted in Fig. 1 using Penrose notation. In the subsequent sections, we denote the matrix obtained as TT with cores $\mathcal{G} = (\mathcal{G}_1, \cdots, \mathcal{G}_k)$ as $\boldsymbol{Y}_{\text{TT}}(\mathcal{G})$ or simply $\boldsymbol{Y}_{\text{TT}}$.

**OOD Detection with TT Point Cloud.** TT point cloud compression is well-suited for OOD detection methods that determine the in-distribution or out-of-distribution status of a query point based on its distance to the nearest neighbor in the normal point cloud (a databank of in-distribution samples). That means, that if the normal point cloud originates from some unknown probability distribution $p_{\text{true}}$, then replacing it with another point cloud that comes from the same (or very close) distribution will not degrade the quality of OOD detection. Therefore, replacing the normal point cloud with TT-compressed point cloud in this setting is straightforward. In comparison to standard techniques where the normal cloud is simply subsampled, either randomly or using methods like coresets, memory gain is achieved due to a more parameter-efficient point cloud representation.

**Approximate Nearest-Neighbour Search.** TT-point clouds cannot be directly applied to the approximate nearest-neighbor search (ANN) problem, as the points contained within them do not

have a direct correspondence with the original vectors. Instead, they are "resampled" from an approximation of the unknown underlying distribution $p_{\text{true}}$. To efficiently address the ANN problem, many approaches first construct an *index* structure on top of the vector database. These index structures serve as sparsifications with a significantly lower number of vectors. All vectors from the database are organized into buckets corresponding to the nearest point from the index structure. For each query vector, one must first find the $K$ nearest neighbors from the index structure and then perform an exhaustive search within their respective buckets.

We propose adapting this approach and using TT-compressed point clouds as an index structure. This approach allows for a much larger in-memory index with better coverage of the vector database. Consequently, this results in more evenly distributed bucket sizes, fewer empty buckets, and overall improved search performance.

One crucial component for this approach to work efficiently is the ability to perform ANN search within the index itself. We demonstrate how this can be achieved for TT point clouds through the utilization of a hierarchical structure implicitly encompassed within in the following section.

**Hierarchical Structure.** Each row index of the matrix $\boldsymbol{Y}_{\text{TT}}$ corresponds to some multiindex $(i_1, \cdots, i_k)$ of the tensor $\boldsymbol{\mathcal{Y}}_{\text{TT}}$. The TT structure enables rapid computation of marginalization along a suffix of indices in the multiindex. Let's define the mean vector of tensor $\boldsymbol{\mathcal{Y}}_{\text{TT}}$ along some suffix of indices $(i_{a+1}, \cdots, i_k)$ as $\boldsymbol{y}^{(a)}_{i_1, \cdots, i_a}$:

$$\boldsymbol{y}^{(a)}_{i_1, \cdots, i_a} = \frac{1}{N_{a+1} \cdots N_k} \sum_{i_{a+1}, \cdots, i_k} \boldsymbol{\mathcal{Y}}_{\text{TT}}[i_1, \cdots, i_k]. \tag{3}$$

The TT representation allows for the efficient evaluation of such sums. Specifically, marginalization along the last index $i_k$ has its own TT structure:

$$\boldsymbol{y}^{(k-1)}_{i_1, \cdots, i_{k-1}} = \boldsymbol{Y}_{\text{TT}}(\boldsymbol{\mathcal{G}}_1, \cdots, \widetilde{\boldsymbol{\mathcal{G}}}_{k-1})[i_1, \cdots, i_{k-1}], \tag{4}$$

$$\widetilde{\boldsymbol{\mathcal{G}}}_{k-1} = \sum_{\alpha=1, i=1}^{r, N_k} \frac{1}{N_k} \boldsymbol{\mathcal{G}}_{k-1}[:, :, \alpha] \boldsymbol{\mathcal{G}}_k[\alpha, i]. \tag{5}$$

Note that the last two cores, $\boldsymbol{\mathcal{G}}_{k-1}$ and $\boldsymbol{\mathcal{G}}_k$, have been contracted together to form a new core, $\widetilde{\boldsymbol{\mathcal{G}}}_{k-1}$. All other cores remain unchanged. The process of marginalization over a suffix of any length can be obtained by induction:

$$\boldsymbol{y}^{(a)}_{i_1, \cdots, i_a} = \boldsymbol{Y}_{\text{TT}}(\boldsymbol{\mathcal{G}}_1, \cdots, \widetilde{\boldsymbol{\mathcal{G}}}_a)[i_1, \cdots, i_a], \tag{6}$$

$$\widetilde{\boldsymbol{\mathcal{G}}}_a = \sum_{\alpha=1, i=1}^{r, N_{a+1}} \frac{1}{N_{a+1}} \boldsymbol{\mathcal{G}}_a[:, :, \alpha] \widetilde{\boldsymbol{\mathcal{G}}}_{a+1}[\alpha, i]. \tag{7}$$

These marginals essentially serve as centroids of clusters, where each cluster is defined by fixing some prefix of indices in $\boldsymbol{Y}_{\text{TT}}$ and together organizes into a hierarchical clusterization structure. It is important to note that there are no guarantees that this clusterization is inherently "good", meaning that each cluster is well-localized and does not intersect with other clusters, however, if it is, efficient approximate nearest-neighbor search can be employed. We consider a beam search-like greedy search approach. For a query vector $\mathbf{q}$, we select the $K$ nearest neighbors from among the cluster centroids at the top level:

$$\mathcal{J}_1 := \left\{ i_{1j} \right\}_{j=1}^K = \text{K-arg} \min_{i_1} \left\| \mathbf{q} - \boldsymbol{y}^{(1)}_{i_1} \right\| \tag{8}$$

Then, in an iterative fashion, we continue to descend in the hierarchical clusterization structure while maintaining a set of the $K$ best candidates for the current centroids level.

$$\mathcal{J}_2 := \left\{ (i_{1j}, i_{2j}) \right\}_{j=1}^K = \text{K-arg} \min_{\substack{i_1, i_2 \\ \text{s.t. } i_1 \in \mathcal{J}_1}} \left\| \mathbf{q} - \boldsymbol{y}^{(a)}_{i_1, i_2} \right\| \tag{9}$$

$$\cdots \tag{10}$$

$$\mathcal{J}_k := \left\{ (i_{1j}, \cdots, i_{kj}) \right\}_{j=1}^K = \text{K-arg} \min_{\substack{i_1, \cdots, i_k \\ \text{s.t. } (i_1, \cdots, i_{k-1}) \in \mathcal{J}_{k-1}}} \left\| \mathbf{q} - \boldsymbol{y}^{(a)}_{i_1, \cdots, i_k} \right\|. \tag{11}$$

The resulting approximate nearest-neighbor is the best candidate among $\mathcal{J}_k$. Both operations, marginalization, and indexing, required for the fast ANN algorithm, can be efficiently implemented for TT in such a way that on each iteration of Eq. (9), the algorithm works with only one core of TT representation. The complete algorithm is described in detail with a code listing in Appendix A.

## 3   LOSSES AND OPTIMIZATION

**Probabilistic Interpretation.**   While there exist efficient methods to obtain a TT approximation with a given rank for a given matrix (such as SVD-based methods Oseledets (2011) and optimization-based methods Oseledets & Tyrtyshnikov (2010); Holtz et al. (2012)), the immediate application of such methods to the matrix $Y$ can yield variable results. This is because rearranging the rows of matrix $Y$ can influence its TT-rank. This implies that for some row orders (enumerations of vectors in the set $\mathcal{Y}$), a more accurate compression of the matrix $Y$ into TT format with a low rank may be achieved, while for other row orders, compression can be highly inaccurate. Notably, the order of rows has no impact on similarity or nearest-neighbor search in the database.

To address the sensitivity of TT approximation to row ordering in the matrix $Y$, we propose a probabilistic interpretation. We consider matrix $Y$ (set $\mathcal{Y}$) as the first point cloud, with elements drawn from a probability distribution $p_Y$. The vectors, encoded in TT format within the matrix $Y_{\mathrm{TT}}$, represent the second point cloud, defining a discrete finite distribution $p_{Y_{\mathrm{TT}}}$. Distribution $p_{Y_{\mathrm{TT}}}$ depends on parameters of TT decomposition – cores $(\mathcal{G}_1, \cdots, \mathcal{G}_k)$. To closely approximate the distribution $p_Y$, we adapt density estimation losses and optimize them using standard SGD methods. In our work, we employ a combination of Sliced Wasserstein Loss Deshpande et al. (2018) and Nearest Neighbor Distance Loss Hennig & Latecki (2003).

**Sliced Wasserstein Loss.**   In our work, we utilize the Sliced Wasserstein Loss to train TT parameters. The Wasserstein Distance, defined between two probability distributions, exhibits many appealing properties, making it suitable for scenarios involving discrete distributions and distributions with different supports, which is particularly important in our case. However, for the general $D$-dimensional case, analytical computation of Wasserstein Distance becomes intractable. In the one-dimensional case, Wasserstein Distance has a simple closed-form solution and can be easily calculated for distributions defined by samples:

$$W_1(\{x_i\}_{i=1}^N, \{y_j\}_{j=1}^N) = \sum_k |x_{i_k} - y_{j_k}|, \tag{12}$$

where $i_k$ and $j_k$ are the indices that sort the sequences $x_i$ and $y_j$, correspondingly:

$$x_{i_k} < x_{i_{k+1}}; \quad y_{j_k} < y_{j_{k+1}}. \tag{13}$$

The Sliced Wasserstein Loss is Wasserstein Distance calculated for random one-dimensional projection of the data. This approach combines the best from both worlds by providing a tractable distance measure for high-dimensional distributions.

$$\mathcal{L}_{\mathrm{SW}}(\{x_i\}_{i=1}^N, \{y_i\}_{j=1}^N) = \mathop{\mathbb{E}}_{u \in U[S^{D-1}]} W_1(\{u^T x_i\}_{i=1}^N, \{u^T y_j\}_{j=1}^N). \tag{14}$$

Here, $\{x_i\}_{i=1}^N$ and $\{y_j\}_{j=1}^N$ are samples from two distributions. $S^{D-1}$ denotes a unit sphere in the D-dimensional space, and $U[S^{D-1}]$ is a uniform distribution on this unit sphere. During training, we estimate the expectation on the right-hand side of Eq. (14) using Monte Carlo estimation.

SW loss works very effectively in the initial optimization stages, capturing the overall similarity in the shapes of two point clouds and achieving strong alignment at a macroscopic level. However, it may struggle to establish precise point-to-point correspondence between the two point clouds at smaller scales. For this reason, we use it in combination with another loss: Nearest Neighbor Distance loss.

**Nearest Neighbour Distance**   loss is a classical loss commonly used in applications like KMeans clustering. In this loss, for each point in the first cloud, we find its nearest neighbor in the second cloud and then summat such distances:

$$\mathcal{L}_{\mathrm{NN}} = \sum_{i=1}^N \min_j \|Y[i] - Y_{\mathrm{TT}}[j]\|^2. \tag{15}$$

In this way, each point in $\boldsymbol{Y}_{\mathrm{TT}}$ moves towards the centroid cluster of its nearest neighbors in $\boldsymbol{Y}$. Since the point cloud $\boldsymbol{Y}$ can be very large, making the computation of $L_{NN}$ computationally intensive, we use an unbiased estimation of it by considering only a random subset of $\boldsymbol{Y}$ (indexed as $i$ in Eq. (15)). We do not subsample $\boldsymbol{Y}_{\mathrm{TT}}$ (indexed as $j$ in Eq. (15)).

Points in $\boldsymbol{Y}_{\mathrm{TT}}$ that do not have any assigned nearest neighbors (corresponding clusters are empty) are not be affected by the loss. While these points are covered by the Sliced Wasserstein loss, we can consider the inverse situation: for each point in $\boldsymbol{Y}_{\mathrm{TT}}$, we find its nearest neighbor among $\boldsymbol{Y}$ and calculate the average of such distances:

$$\mathcal{L}_{\mathrm{NN}}^{\mathrm{inv}} = \sum_{j=1}^{N} \min_{i} \|\boldsymbol{Y}[i] - \boldsymbol{Y}_{\mathrm{TT}}[j]\|^2 \tag{16}$$

and take linear combination of this two losses with some coefficient $\alpha$:

$$\mathcal{L}_{\mathrm{NN}}^{\mathrm{total}} = \alpha \mathcal{L}_{\mathrm{NN}} + (1 - \alpha)\mathcal{L}_{\mathrm{NN}}^{\mathrm{inv}}. \tag{17}$$

## 4 EXPERIMENTS

### 4.1 TOY EXAMPLES

To provide some intuition about TT point clouds, consider the toy examples shown in Fig. 2. Each toy point cloud consists of 8192 vectors, and we compress it with a TT representation consisting of two cores with sample dimensions 64 and 128 and a TT-rank of 8. For each toy point cloud, you can observe the centroids of the first and second levels (see Eq. (3)) in the first and second columns, respectively. In the third column of Fig. 2, you can see the probability density of the original point cloud, and in the fourth column, the probability density of the TT point cloud. The trained vectors reconstruct the original point clouds very well, capturing all important details. They are able to approximate both one- and two-dimensional manifolds: the first point cloud consists of a one-dimensional semicircle, the second point cloud consists of 16 one-dimensional circles arranged in a two-dimensional grid, and the third point cloud consists of three sub-clouds: one with a uniform distribution, one with a normal distribution, and one with a one-dimensional curve featuring a non-uniform distribution (see details in Appendix B).

### 4.2 MVTEC DATASET

MVTec AD Bergmann et al. (2019) is an anomaly detection benchmark that contains 15 different datasets. Each contains training set of anomaly-free images and a test set that contains both images with and without anomalies. Each anomaly has a mask, so per-pixel metrics can be evaluated.

To reduce the memory consumption of the feature databank, the PatchCore method employs coreset subsampling, which greedily solves the following optimization problem over subsampled indices $\mathcal{I} = \{i_k\}_{k=1}^{N'}$ for databank $\mathcal{Y} = \{\boldsymbol{y}_i\}$:

$$\arg \min_{\mathcal{I}} \left\{ \max_{\boldsymbol{y} \in \mathcal{Y}} \min_{i \in \mathcal{I}} \|\boldsymbol{y} - \boldsymbol{y}_i\| \right\}.$$

Here, $N' = cN$, where $N$ is the size of the original feature databank, and $c$ is a subsampling factor. Instead of using coreset subsampling, we compress the point cloud $\mathcal{Y}$ into a TT Point Cloud $\boldsymbol{Y}_{\mathrm{TT}}$. We use TT representation with two cores: the first one of size $1024 \times N_1 \times r$ and the second of size $r \times N_2$. In all our experiments, for a given subsample factor $c$, we choose hyperparameters for the TT point cloud, namely the number of sample factors $N_1$ and $N_2$ and the rank $r$, so that the total number of parameters in the TT $(1024N_1r + N_2r)$ equals to the total parameters in the coreset-subsampled feature databank $(1024cN)$. We choose the same hyperparameters for all MVTec datasets to avoid possible hyperparameter overfitting and to demonstrate the stability of our method, which can be highly desirable in real-world use cases where the method may be applied to related but previously unseen domains.

The standard comparison metrics include pixel-level and image-level AUROC. While we report these values, we found that they are not sufficiently representative. For most of the subdatasets in the

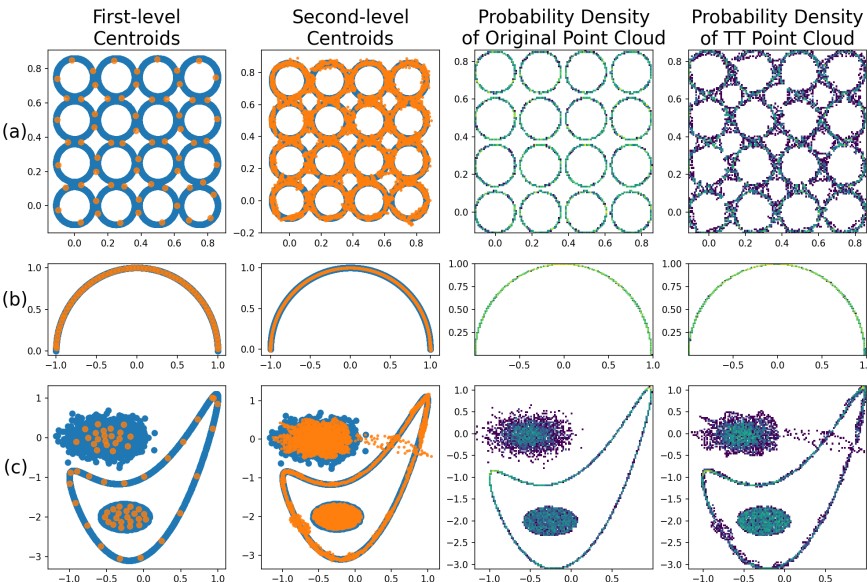

Figure 2: Three toy point clouds (blue points) consisting of 8192 vectors each, and its compressed TT-point cloud approximation (orange points).

MVTec benchmark, the AUROC values are extremely close to 1, often exceeding 0.99. Instead, we also report the area under the precision-recall curve and a specific value, $P@R90$, which represents precision when recall is 90%. This metric is much more challenging for the MVTec benchmark, as some datasets show $P@R90$ values as low as 0.19 for the full (not subsampled) PatchCore feature databank.

In Table 1, we present results for 1% subsampling (corresponding to a 100x compression ratio) and 0.1% subsampling (corresponding to a 1000x compression ratio). All pixel-level metrics heavily favor TT Point Cloud, and as the compression ratio increases, the gap only widens. Image-level metrics are quite competitive with coreset subsampling, with TT Point Cloud showing a slight advantage.

### 4.3 APPROXIMATE NEAREST NEIGHBORS

ANN is a very well-developed and competitive field, with highly optimized solutions, both algorithmically and in terms of implementation. There are many state-of-the-art libraries and frameworks like FAISS git (a) or NGT git (b), which have undergone extensive low-level optimizations. We acknowledge that it is challenging to compete with such high-end solutions. Instead, we propose a proof-of-concept solution for using TT point cloud as an index structure. In our evaluation, we focus on indirect characteristics such as the quality of dataset coverage, rather than providing actual queries-per-second values.

We conduct our experiments on the Deep1B Babenko & Lempitsky (2016) – dataset, that contains one billion 96-dimensional vectors, that was produced as an outputs from last fully-connected layer of ImageNet-pretrained GoogLeNet Szegedy et al. (2015) model, then compressed by PCA to 96 dimensions and $l_2$-normalized. In our experiments we use only subset of size $10M$ vectors.

We build proof-of-concept ANN system based on TT point cloud, where we use TT point cloud as an indexing tree structure, which plays a role of first-stage database filtration. During training step, each vector in the database is distributed to the bucket of the closest to it point from TT point cloud. During inference, for the query vector $\mathbf{q}$ we first search for $K$ nearest points from TT point cloud, using efficient hierarchical search technique, described in Section 2. Then look for nearest neighbour only inside short-list, that consists of the corresponding buckets.

(a) ×100 compression ratio.

| dataset | pixel p@r=90 ttm | pixel p@r=90 coreset | pixel AUROC ttm | pixel AUROC coreset | pixel AUPRC ttm | pixel AUPRC coreset | image p@r=90 ttm | image p@r=90 coreset | image AU-ROC ttm | image AU-ROC coreset | image AUPRC ttm | image AUPRC coreset |
|---|---|---|---|---|---|---|---|---|---|---|---|---|
| Bottle | *0.780* | 0.777 | *0.990* | 0.988 | *0.904* | 0.901 | *1.000* | *1.000* | *1.000* | *1.000* | *1.000* | *1.000* |
| Cable | *0.545* | 0.396 | *0.986* | 0.971 | *0.830* | 0.792 | *0.993* | 0.991 | *0.995* | 0.994 | *0.997* | 0.996 |
| Capsule | *0.366* | 0.339 | *0.988* | 0.979 | *0.661* | 0.646 | 0.993 | *0.994* | *0.990* | 0.988 | *0.998* | 0.998 |
| Carpet | *0.496* | 0.474 | *0.986* | 0.985 | *0.805* | 0.802 | *1.000* | 0.995 | *0.982* | 0.981 | *0.995* | 0.994 |
| Grid | *0.398* | 0.294 | *0.986* | 0.976 | *0.757* | 0.734 | *1.000* | 1.000 | 0.987 | *0.990* | 0.996 | *0.997* |
| Hazelnut | *0.634* | 0.610 | *0.989* | 0.987 | *0.843* | 0.834 | *1.000* | 1.000 | *1.000* | *1.000* | *1.000* | 1.000 |
| Leather | 0.478 | *0.479* | *0.995* | 0.995 | *0.735* | 0.734 | *1.000* | 1.000 | *1.000* | 1.000 | *1.000* | 1.000 |
| Metal Nut | *0.809* | 0.710 | *0.981* | 0.965 | *0.929* | 0.909 | *1.000* | *1.000* | *0.999* | 0.998 | *1.000* | 1.000 |
| Pill | *0.365* | 0.265 | *0.967* | 0.952 | *0.810* | 0.792 | *0.991* | 0.990 | 0.975 | *0.975* | 0.996 | *0.996* |
| Screw | *0.155* | 0.075 | *0.984* | 0.961 | 0.562 | *0.574* | 0.959 | *0.963* | 0.952 | *0.962* | 0.979 | *0.985* |
| Tile | *0.645* | 0.641 | *0.972* | 0.970 | 0.801 | *0.810* | *1.000* | 1.000 | *1.000* | 1.000 | *1.000* | 1.000 |
| Toothbrush | *0.541* | 0.459 | *0.991* | 0.981 | *0.812* | 0.785 | *0.923* | 0.908 | *0.929* | 0.902 | *0.967* | 0.953 |
| Transistor | *0.260* | 0.135 | *0.949* | 0.896 | *0.673* | 0.630 | *1.000* | 0.997 | 0.993 | *0.994* | 0.991 | *0.992* |
| Wood | *0.386* | 0.367 | *0.956* | 0.953 | *0.706* | 0.706 | *1.000* | 0.999 | 0.988 | *0.989* | 0.996 | *0.997* |
| Zipper | *0.520* | 0.512 | *0.987* | 0.985 | 0.784 | *0.786* | *1.000* | 0.996 | *0.996* | 0.993 | *0.999* | 0.998 |
| Mean | *0.492* | 0.435 | *0.980* | 0.970 | *0.774* | 0.762 | *0.991* | 0.989 | *0.986* | 0.984 | *0.994* | 0.994 |

(b) ×1000 compression ratio.

| dataset | pixel p@r=90 ttm | pixel p@r=90 coreset | pixel AUROC ttm | pixel AUROC coreset | pixel AUPRC ttm | pixel AUPRC coreset | image p@r=90 ttm | image p@r=90 coreset | image AU-ROC ttm | image AU-ROC coreset | image AUPRC ttm | image AUPRC coreset |
|---|---|---|---|---|---|---|---|---|---|---|---|---|
| Bottle | *0.755* | 0.627 | *0.989* | 0.976 | *0.896* | 0.864 | *1.000* | *1.000* | *1.000* | 1.000 | *1.000* | 1.000 |
| Cable | *0.375* | 0.151 | *0.976* | 0.925 | *0.769* | 0.641 | *0.941* | 0.913 | *0.969* | 0.955 | *0.982* | 0.975 |
| Capsule | *0.280* | 0.065 | *0.984* | 0.927 | *0.618* | 0.538 | 0.969 | *0.973* | 0.956 | *0.958* | 0.990 | *0.992* |
| Carpet | *0.478* | 0.428 | *0.986* | 0.981 | *0.805* | 0.792 | *0.998* | 0.991 | *0.981* | 0.978 | *0.995* | 0.994 |
| Grid | *0.259* | 0.092 | *0.978* | 0.938 | *0.712* | 0.625 | 0.970 | *0.981* | *0.976* | 0.970 | *0.992* | 0.990 |
| Hazelnut | *0.572* | 0.415 | *0.986* | 0.979 | *0.830* | 0.786 | *1.000* | 0.998 | *1.000* | 0.999 | *1.000* | 1.000 |
| Leather | *0.489* | 0.468 | *0.995* | 0.994 | *0.751* | 0.723 | *1.000* | 1.000 | *1.000* | 1.000 | *1.000* | 1.000 |
| Metal Nut | *0.644* | 0.400 | *0.970* | 0.933 | *0.900* | 0.818 | *1.000* | 0.996 | *0.992* | 0.976 | *0.998* | 0.995 |
| Pill | *0.314* | 0.136 | *0.962* | 0.920 | *0.785* | 0.708 | *0.976* | 0.965 | *0.959* | 0.946 | *0.993* | 0.990 |
| Screw | *0.064* | 0.028 | *0.962* | 0.913 | *0.298* | 0.252 | 0.792 | *0.800* | 0.735 | *0.774* | 0.894 | *0.916* |
| Tile | *0.635* | 0.583 | *0.970* | 0.962 | *0.797* | 0.792 | *1.000* | 1.000 | *1.000* | 0.999 | *1.000* | 1.000 |
| Toothbrush | *0.350* | 0.165 | *0.983* | 0.956 | *0.777* | 0.661 | *0.902* | 0.792 | *0.918* | 0.816 | *0.964* | 0.931 |
| Transistor | *0.156* | 0.083 | *0.903* | 0.805 | *0.587* | 0.495 | *0.977* | 0.961 | 0.956 | *0.969* | 0.965 | *0.971* |
| Wood | *0.390* | 0.335 | *0.957* | 0.948 | *0.709* | 0.692 | *1.000* | 1.000 | 0.989 | *0.990* | 0.997 | *0.997* |
| Zipper | *0.457* | 0.310 | *0.984* | 0.968 | *0.751* | 0.734 | 0.984 | *0.999* | 0.986 | *0.993* | 0.996 | *0.998* |
| Mean | *0.415* | 0.286 | *0.972* | 0.942 | *0.732* | 0.675 | *0.967* | 0.958 | *0.961* | 0.955 | *0.984* | 0.983 |

Table 1: Metrics for the MVTec AD benchmark comparing TT point cloud-based feature databank subsampling and coreset-based subsampling for Patchcore feature vectors. AUROC represents the area under the ROC curve, AUPRC is the area under the precision-recall curve, and $P@R90$ is the precision at recall $= 90\%$. Values are averaged across 16 runs.

We compare with similar technique (G)NO-IMI Babenko & Lempitsky (2016). It uses the same idea with two-level indexing tree structure of the form

$$\{\mathbf{S}_i + \alpha_{i,j}\mathbf{T}_j\}, \tag{18}$$

where $\mathbf{S}_i$ and $\mathbf{T}_j$ represent two sets of vectors, first and second-level, respectively. $\alpha$ is an additional weight matrix that is used only in GNO-IMI modification (for NO-IMI $\alpha_{i,j} = 1\forall i, j$), that adds more flexibility for the construction with the cost of more memory. In the greedy search approach of GNO-IMI, the algorithm initially seeks the top $K$ candidates among $\mathbf{S}_i$ and subsequently explores the best $K$ candidates among $\mathbf{S}_i + \alpha_{i,j}\mathbf{T}_j$ for the indices $i$ determined in the preceding step. In their work Babenko & Lempitsky (2016), it is suggested to utilize sets $\{\mathbf{S}_i\}$ and $\{\mathbf{T}_j\}$ of equal size $2^{14}$. However, given that we are working with a subset of the Deep1B dataset, we have adjusted this quantity to $2^{10}$. This adaptation results in a total of $2^{20}$ vectors in the index structure, encompassing $2 \cdot 96 \cdot 2^{10} + 2^{20} = 1.2$ million parameters.

We compare this structure with a TT point cloud containing three cores, with sample dimensions of 64, 64, and 256, resulting in the same number of buckets. To match the memory consumption of the TT point cloud and GNO-IMI, we can use a TT-rank as large as 96.

The complexity of ANN search with an indexing structure consists of two factors. First, how fast and efficiently we can search for bucket candidates, and second, how good these buckets are. Even if the indexing structure somehow manages to store many vectors using a small amount of memory, if most of the corresponding buckets are empty and only a few of them contain the full vector database, then the second stage of the search will still be required to exhaustively search through a large proportion of the database. If the indexing structure has $M$ buckets, with $i$-th bucket of size $N_i$, then the minimum number of points that need to be searched to find the nearest neighbor of vector $\mathbf{q}$ is given by:

$$\sum_{i=1}^{M} p_i N_i, \tag{19}$$

where $p_i$ is the probability that the nearest neighbor of $\mathbf{q}$ is among the $i$-th bucket. We can estimate $p_i$ based on the size of the $i$-th bucket itself as $\frac{N_i}{N}$, where $N = \sum_{i=1}^{M} N_i$ is the total number of vectors

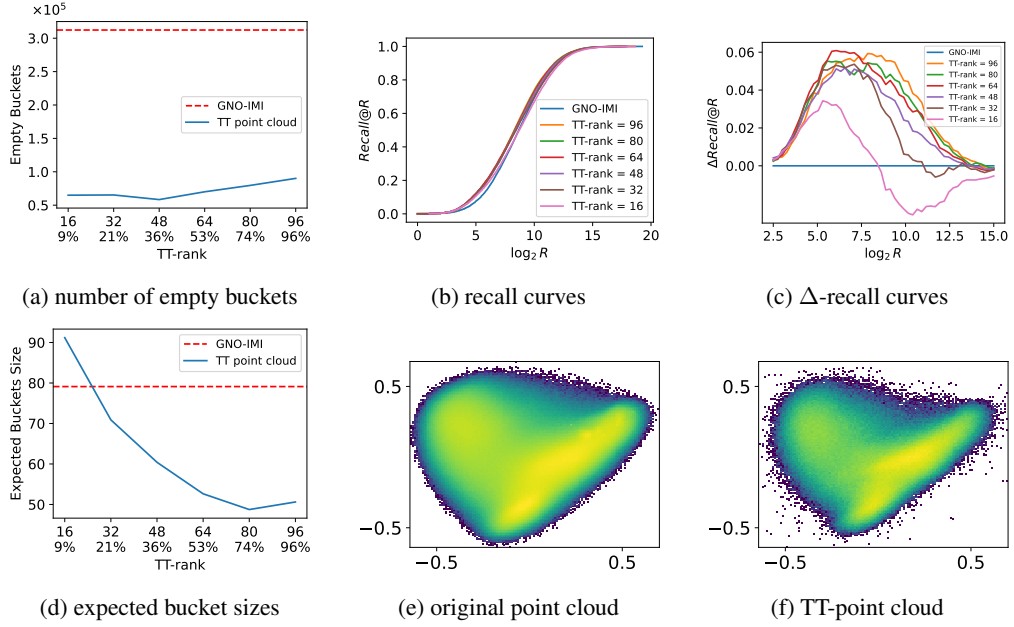

Figure 3: (a, b, c, d): Comparison of different characteristics of the index search structure for GNO-IMI and TT-point cloud with varying TT-ranks. On the x-axis, we plot TT-rank (upper value) and the fraction of parameters of the TT point cloud relative to the number of parameters in GNO-IMI (bottom value). (e, f): Two-dimensional PCA projections of the original point cloud (e) and TT-point cloud (f).

in the database. Then, the expected number of points to be searched through is:

$$\sum_{i=1}^{M} p_i N_i = \sum_{i=1}^{M} \frac{N_i}{N} N_i \propto \sum_{i=1}^{M} N_i^2 \qquad (20)$$

On Figs. 3e and 3f you can see two-dimensional PCA projections of the original point cloud (which has shape $10\text{M} \times 96$) and TT-compressed point cloud (representing point cloud of size $1M \times 96$ but taking only 671K parameters).

Fig. 3d compares dependence of Eq. (20) and Fig. 3a compares number of empty buckets for GNO-IMI and TT with different ranks. TT point cloud has significantly less empty buckets: almost $\times 6$ less for all ranks. Expected bucket size Eq. (20) is lower already with TT-rank $= 32$ (only 21% of GNO-IMI size).

Fig. 3b illustrates $Recall@R$ for different values of $R$ – rate of queries for which true nearest neighbor is present in a short-list of length $R$. As the curves are quite similar, we built a plot of curve difference with GNO-IMI as a baseline: we plot original $Recall@R$ curve minus GNO-IMI $Recall@R$ curve on Fig. 3c. All TT-ranks starts with quite high advantage over GNO-IMI and the larger the rank – the longer and bigger this advantage continues. Although at the end for recalls very close to one there is a small period of time when GNO-IMI is better.

# 5 RELATED WORK

TT Point Cloud is built upon tensor-train low-rank decomposition, which has a rich history of applications in Machine Learning and Deep Learning. Tensor-Train Matrix decomposition, a specialized form of TT decomposition, shares common principles with our decomposition approach. It has been employed to compress large weight matrices of linear classification layers in fully-convolutional networks Novikov et al. (2015), vocabulary embedding layers in language models Hrinchuk et al. (2020), and linear layers in transformer-based networks Chekalina et al. (2023) and others Phan

et al. (2020). TT decomposition has demonstrated its efficiency in parameterizing high-dimensional distributions Novikov et al. (2021); Kuznetsov et al. (2019).

Coreset subsampling Agarwal et al. is a well-established technique with applications in various contexts. In essence, a coreset is a subset $\mathcal{S} \subset \mathcal{A}$ that allows for a good approximation of the solution to a problem over the entire set $\mathcal{A}$.

Patchcore Roth et al. (2022) extracts pixel-level features from a set of normal images and make a decision whether a query point belongs to the in-distribution or out-of-distribution class based on the distance to the nearest neighbour among this databank. To reduce memory consumption of the databank and speed up NN search, Patchcore utilizes a specific type of coreset that is particularly well-suited for optimizing nearest-neighbor search performance Sener & Savarese (2018). Other methods that use pixel-level feature databank are SPADE Cohen & Hoshen (2021) and PaDiM Defard et al. (2020). But they do not use the full set of vectors during inference, as SPADE incorporates a distinct initial image-level filtering procedure to operate with relatively small sets of feature vectors, and PaDiM learns a parametric probabilistic representation of the normal class. Notably, Patchcore exhibits significant improvements over similar methods, underscoring the effectiveness of using the full feature databank and the importance of its efficient representation.

ANN search is a crucial problem with various solutions, employing different techniques, including locally-sensitive hashing Jafari et al. (2021), graph-based approaches Wang et al. (2021). IVFADC Jégou et al. (2011) constructs a one-level indexing tree structure, effectively performing k-means clustering. IMI Babenko & Lempitsky (2015) utilizes product quantization with two components instead of vector quantization for the bucket centroids, resulting in an index of quadratic size when compared to the IVFADC approach. (G)NO-IMI Babenko & Lempitsky (2016) builds upon this concept and develops a two-level indexing structure with the capability of approximate nearest neighbor search within it, achieving a significantly better balance between retrieval speed and recall.

## 6 LIMITATIONS

While TT Point Cloud demonstrates an exellent ability to represent point clouds, one notable limitation of this approach is its inability to establish a direct one-to-one correspondence between points from the original point cloud and points from the TT point cloud. This constraint restricts its applicability to situations where the primary focus is on the underlying distribution rather than the original point cloud itself. For instance, TT Point Cloud is highly suitable for solving out-of-distribution (OOD) problems based on feature databanks and cannot be employed directly for nearest-neighbor search tasks. Nevertheless, we have shown how TT Point Cloud can still be utilized effectively in such scenarios as a first-stage search index structure.

## 7 CONCLUSION

The use of large vector databases in modern machine learning raises the question of their efficient storage and fast vector retrieval. We propose using tensor-train low-rank tensor decomposition to represent large point clouds with a small number of parameters. We introduce a probabilistic interpretation of point cloud approximation to achieve a compression method that is invariant to vector reordering in point cloud. Minimization of Sliced Wasserstein Eq. (14) and Nearest-Neighbour Distance Loss Eq. (17) allows to interpret points from TT point cloud as a newly generated points, sampled from the same distribution as points in the original point cloud. TT point clouds are particularly suitable for methods interested in the underlying distribution rather than the vectors themselves, such as out-of-distribution detection based on feature databanks. We tested the proposed OOD feature databank compression method and compared it to coreset subsampling on MVTec AD benchmark. In the crucial area of approximate nearest-neighbor search, where TT point clouds cannot be directly applied, we proposed a proof-of-concept solution: using TT Point Cloud as an indexing structure for the initial filtering of a small number of vectors, in which the nearest neighbors can then be searched through by exhaustive search. We demonstrated the advantages of this approach on the Deep1B dataset in comparison to another indexing method, GNO-IMI. All experiments and implementation of the proposed method can be found in `github.com:/omitted/for/anonymity`.

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

## A   BEAM SEARCH

---

**Algorithm 1:** Beam search ANN for TT-point cloud

---
**Data:** Parameters of TT-point cloud $\mathcal{G}_1, \cdots, \mathcal{G}_k$. Precomputed marginalization cores $\widetilde{\mathcal{G}}_1, \cdots, \widetilde{\mathcal{G}}_k$ Eq. (6). Queary vector $\mathbf{q}$. Parameter $K$ that determines how many nearest neighbors should be found.

**Result:** Set $\mathcal{J} = \left\{ (i_{1j}, \cdots, i_{kj}) \right\}_{j=1}^{K}$ of approximate nearest neighbors of vector $\mathbf{q}$ among vectors of $\boldsymbol{\mathcal{Y}}_{\mathrm{TT}}$

1  Take first-level centroids $\boldsymbol{y}_{i_1}^{(1)} = \widetilde{\mathcal{G}}_1[i_1, :]$ ;

2  Calculate distances between $\mathbf{q}$ and $\boldsymbol{y}_{i_1}^{(1)} : d_i = \left\| \boldsymbol{y}_i^{(1)} - \mathbf{q} \right\|^2$ ;

3  Take indices of $K$ closest first-level centroids: $\mathcal{J}_1 \leftarrow \left\{ i_{1j} \right\}_{j=1}^{K} = \underset{i}{\mathrm{K\text{-}arg\,min}}\, d_i$ ;

4  Take $K$ slices of first TT-core $\mathcal{G}_1$ to be able to efficiently calculate higher-level centroids with fixed prefix of indices: $\mathbf{L}_j \leftarrow \mathcal{G}_1[:, i_{1j}, :] \forall j = 1..K$;

5  **for** *current centroids level* $l \leftarrow 2$ **to** $k$ **do**

6       Evaluate $l$-th level centroids with indices from $\mathcal{J}_{l-1}$: $\boldsymbol{y}_{j,i_l} \leftarrow \boldsymbol{y}_{i_{1j}, \cdots, i_{l-1j}, i_l}^{(l)} = \mathbf{L}_{:,j,:} \widetilde{\mathcal{G}}_l[:, i_l]$ ;

7       Calculate distances between $\mathbf{q}$ and $\boldsymbol{y}_{j,i_l} : d_{j,i_l} \leftarrow \| \boldsymbol{y}_{j,i_l} - \mathbf{q} \|^2$ ;

8       Take top-K best candidates: $\mathcal{J}' = \{ (j_a, i_{la}) \} = \underset{j,i_l}{\mathrm{K\text{-}arg\,min}}\, d_{j,i_l}$ ;

9       Update set with indices of current top-$K$ candidates: $\mathcal{J}_l = \left\{ (i_{1j}, \cdots, i_l) | (j, i_l) \in \mathcal{J}' \right\}$ ;

10       Update state matrices $\mathbf{L}_1, \cdots, \mathbf{L}_K$: $\mathbf{L}_a \leftarrow \mathbf{L}_{j_a} \mathcal{G}_l[:, i_{la}, :]$ s.t. $(j_a, i_{la}) \in \mathcal{J}'$ ;

11  **return** $\mathcal{J}_k$;

---

Efficient implementation of beam search over TT point cloud consists of two components:

1. Fast and efficient marginals calculation to be able to obtain centroids of various levels

2. Fast and efficient maintaining of such centroids, indices of which are already chosen by beam search.

For the first component, we need to precalculate surrogate TT-cores $\widetilde{\mathcal{G}}_1, \cdots, \widetilde{\mathcal{G}}_k$ as described in Eq. (6). This has to be done only once and then can be reused for any number of queries.

For the second component, consider calculating $\boldsymbol{y}_{i_1, \cdots, i_l}^{(l)}$ for fixed $i_1, \cdots, i_{l-1}$ and all possible $i_l$:

$$\boldsymbol{y}_{i_1, \cdots, i_l}^{(a)}[d] = \boldsymbol{Y}_{\mathrm{TT}}(\mathcal{G}_1, \cdots, \widetilde{\mathcal{G}}_l)[i_1, \cdots, i_l] \tag{21}$$

$$= \mathcal{G}_1[d, i_1, :]\mathcal{G}_2[:, i_2, :]\cdots \widetilde{\mathcal{G}}_l[:, i_l] \tag{22}$$

$$= \mathbf{L}[d, :]\widetilde{\mathcal{G}}_l[:, i_l], \tag{23}$$

where

$$\mathbf{L}[d, :] = \mathcal{G}_1[d, i_1, :]\cdots \mathcal{G}_{l-1}[:, i_2, :]. \tag{24}$$

In line 1 we maintain all $K$ such auxiliary matrices $\mathbf{L}_1, \cdots, \mathbf{L}_L$: first they are initialized in line 4 and updated on each for-loop iteration in line 10.

Complexity of the beam search is as follows:

1. For the first-level centroids we need to calculate distances in line 2 in $\mathcal{O}(DN_1)$ time and then initialize matrices $\mathbf{L}$ in line 4 in $\mathcal{O}(KDr)$ time.

2. For each other level centroids, we need to calculate centroids in line 6 in $\mathcal{O}(kdrn)$ time, where $n = \max_{i=2..k-1} N_i$. Then update matrices $\mathbf{L}$ in line 10 in $\mathcal{O}(KDr^2)$ time.

The total asymptotic of Alg.1 is $\mathcal{O}(DN_1 + kKDrn + kKDr^2)$ time.

## B  EXPERIMENT SETUPS

### B.1  TOY EXAMPLES

First toy point cloud is 16 uniform distributions over circles, arranged in 4-by-4 grid. Each circle is chosen uniformly.

Second toy point cloud is a uniform distribution over one half of a circle.

Thirds toy point cloud is a mixture of three distributions with equal weights:

1. non-uniform distribution over 1-dimensional closed loop
2. uniform distribution over disc (blob inside closed loop)
3. normal distribution (blob above closed loop)

### B.2  MVTEC AD

For all datasets in MVTec AD benchmark we used same hyperparameters.

We followed original implementation[1] of Patchcore for feature extraction and coreset subsampling. We trained TT point cloud via riemannian stochastic gradient descent Novikov et al. (2022) for $2^{13}$ iterations with initial learning rate equals to $10^3$ and exponential decay of $\frac{1}{3}$ every 256 iterations. Before multiplying by learning rate, we scale gradient to 1 to achieve some kind of adaptivity for riemannian SGD.

Parameters of the loss function:

1. SW loss with coefficient 1 and batch size 32
2. NN Distance loss with $\alpha = 0.1$ and coefficient 0.1, with random $2^{11}$ samples from $Y$ on each iteration.

### B.3  APPROXIMATE NEAREST NEIGHBOUR

To train TT point cloud as an ANN indexing structure we performed two-stage optimization. On the first stage we used Riemannian SGD Novikov et al. (2022) with combination of SW loss with coefficient 1 and batch size 32, and NN Distance loss with $\alpha = 0.1$ and coefficient 0.1 with random $2^{15}$ samples from $Y$ on each iteration. Initial learning rate $10^2$ with exponential decay of $\frac{1}{3}$ every $2^{11}$ iterations. Number of iterations $2^{13}$.

On the second stage we used ALS Holtz et al. (2012) with only NN Distance loss with $\alpha$ equals to 0.001 for 256 iterations.

---

[1]https://github.com/amazon-science/patchcore-inspection

