# OpenReview forum: "Tensor-Train Point Cloud Compression and Efficient Approximate Nearest Neighbor Search"
_ICLR.cc/2024/Conference — Submitted to ICLR 2024_

### Official Review · Reviewer_KPPr · 2023-10-26

**Soundness:** 3 good
**Presentation:** 1 poor
**Contribution:** 2 fair
**Rating:** 3
**Confidence:** 4

**Summary:**

In this work, the authors proposed a method to represent point cloud data into the tensor train (TT) format. Furthermore, they studied its applications to out-of-distribution (OOD) detection and the approximate nearest-neighbor(ANN) search task.

**Strengths:**

This paper discussed several insightful uses of TT, like using TT to model point clouds and using the TT format for efficient K-means. In addition, it is also interesting to use Wasserstein loss for learning the TT cores.

**Weaknesses:**

1. The presentation of this paper is poor. Maybe the authors assume that the writers are familiar with all the details about the point cloud and tensor. But it’s not true at least on my side. The paper involves many specific concepts but these concepts are not explained clearly. In addition, the organization of the content is also bad. It is hard for me to follow the main flow of this work.
2. Although several interesting new ideas are mentioned in the paper, the discussion is not sufficient to convince the writer to agree with the superior performance of using TT in the point cloud task.

**Questions:**

1. In Eq. (1), may I know the specific definition of the operation reshape( . )?
2. Eq (8) seems unclear. Could you explain more about it?
3. On page 2, you mentioned, “replacing the normal point cloud with TT-compressed point cloud in this setting is straightforward”. Could you explain more why it is straightforward?
4. I’m still not clear on how to relate TT to different levels in the clustering task. Please give more intuition if possible.
5. In the experiments, it seems that only 2~3rd-order TTs are used. Have you ever considered using higher orders? Do any potential issues exist in the higher-order case?

---

> ### Author Response · Authors · 2023-11-23
>
> First of all, Thank you for insightful comments and constructive feedback. I truly believe that your suggestions will contribute significantly to making this work better!
>
>
>
> 1. Operator reshaping resembles similar operators from the numpy or torch.
> In Equation 8, we choose $K$ closest first-level centroids. $q$ here is a query vector, and $y_i$ are different first-level centroids. operator K-argmin will choose such $K$ indices $i_1, i_2, …, i_K$, that corresponding vectors $y_{i_1}, …, y_{i_k}$ are closest to $q$ among all $y_i$ and the resulting set of indices is stored in $J_1$. Then, in the following equation, the same procedure repeats with the second-level centroids, but instead of using all second-level centroids, only those with the first index from $J_1$ are used. Because of this filtration, it is not an exhaustive search but a fast, approximate search.
>
> 2. By "straightforward," we meant that we could just use an approximate, compressed point cloud in place of the original point cloud. So previously, the OOD decision was performed as follows: for query vector q, calculate the distance to the closest point in the training point cloud. If this distance is greater than some threshold, q is an out-of-distribution sample. Otherwise, it is a normal sample. If the approximated compressed point cloud is perfect, then it is sampled from the same distribution as the original point cloud, and thus we can just calculate the distance from $q$ to the compressed point cloud.
>
>     This is in contrast with ANN, where we cannot use even perfectly trained compressed point clouds. As it does not contain vectors from the original database, the nearest neighbor point from the compressed point cloud will be a meaningless, newly generated vector without any correspondence with the original database.
>
> 3. If the TT point cloud represents a tensor of shape $[N_1, N_2, …, N_k, D]$, then first-level centroids are arranged in a tensor of shape $[N_1, D]$ that is a mean value across $N_2, … N_k$ dimensions. Second-level centroids are arranged in tensor of shape $[N_1, N_2, D]$ that are mean values across $N_3, … N_k$. And so on.
>
> 4. We can use these centroids in algorithms because the TT structure of our point cloud allows for a very fast calculation of any level centroid with any indices without explicitly instantiating any tensors in memory. And that is used in our algorithms to achieve good asymptotics.
>
> 5. Usage of higher-order TTs is associated with the following difficulties: first of all, more cores in TT will give a significant improvement only in larger-scale point clouds. Second of all, surely the more cores in the TT representation, the more unstable and hard-to-train the whole construction may become. But in this paper, we did not have the computational ability to process larger datasets (for example, we performed our experiments only on the 10M subset of the Deep1B dataset). And the point clouds in the MVTecAD benchmark are even smaller. So there was no need to use more than 3 cores.
> But nevertheless, it is possible to use higher-order TTs, and this is an important future work direction.

---

### Official Review · Reviewer_xaNX · 2023-11-01

**Soundness:** 2 fair
**Presentation:** 3 good
**Contribution:** 1 poor
**Rating:** 3
**Confidence:** 4

**Summary:**

This paper proposes training a tensor decomposition of a dataset for use in anomaly detection and nearest neighbor search. The tensor decomposition is trained on a combination of the Sliced Wasserstein Loss, which is the Wasserstein loss applied to a random 1d projection of the data, and a subsampled nearest neighbors loss, where for the squared L2 distance from a random subset of the data to each respective nearest neighbor in the resulting tensor point cloud is added to the loss.

The resulting tensor decomposition makes it easy to generate points that should lie on a roughly similar distribution as the original dataset, which can be used for anomaly detection. The points from the tensor decomposition may also be used as the basis for an inverted index style data structure that can accelerate approximate nearest neighbor search.

**Strengths:**

1. Tensor decomposition should indeed be much smaller than the original dataset.
1. The tensor decomposition is amenable to random sampling.
1. Tensor decomposition background, ANN beam search, and training loss are well-delineated.

**Weaknesses:**

1. No mention of training time: without this key metric, benchmarks are rather meaningless. For example, for nearest neighbor search, running Lloyd's algorithm on the 10M vectors to generate ~12K clusters (also around 1.2M parameters, the same number) will lead to almost no empty clusters, and much better recall (3b, 3c) than either approach. However, it may also be more computationally expensive. Without documenting training time, the quality-compute tradeoff is not made clear.
1. Weak results on nearest neighbor search. Comparison is only done against GNO-IMI, but at the 10M vector scale, it should be fairly easy to compare with many inverted index style data structures. Notably, FAISS/ScaNN both use k-means clustering, which would naturally fit onto Figure 3b. GNO-IMI is a weak baseline that should at best be used at larger scales when training costs must be kept to an absolute minimum. The paper also alternates between proposing tensor decomposition as an interesting proof of concept for nearest neighbors (if so, what future research or experiments must be done to see if this is truly a viable candidate for efficient nearest neighbor search?) and  as something that's already effective and should be used immediately (which this paper fails to convince me of).
1. Figures 3e and 3f: PCA-projecting 96 dimensions onto two dimensions is not difficult, and even the most naive compression methods will result in something that looks similar once projected to two dimensions, so these plots aren't very significant.
1. Figure 2 is confusing; for example, in 2a, why are there so few orange points on the left column, relative to 2b or 2c? Are the same points overlaid over the first two columns? What is the significance of the color differences in the last two columns?

**Questions:**

How does $\alpha=0$ or $\alpha=1$ (i.e., either the Sliced Wasserstein Loss alone, or the nearest neighbor loss alone) perform?

---

> ### Author Response · Authors · 2023-11-23
>
> Thank you for insightful comments and constructive feedback. I truly believe that your suggestions will contribute significantly to making this work better.
>
> Sliced Wasserstein loss alone is good enough to approximately fit the cloud in terms of overall shape, scale, location, and form. So visually, the two-dimensional PCA projections of the training cloud and the approximated cloud are very similar. But it has very poor actual task-dependent quality metrics (like quality of OOD detection or recall for ANN search). That means that Sliced Wasserstein cannot achieve precision on smaller, point-to-point scales at the end of the optimization process on its own.
>
> Nearest Neighbor Loss, on the other hand, cannot be applied alone from the beginning of the optimization, as the nearest neighbor correspondence would be degenerate: only a few points from the TT point cloud will have non-empty clusters. But when used together with Sliced Wasserstein in the latter phases of the training, when two clouds are already very similar, Nearest Neighbor loss makes up for the lack of Sliced Wasserstein precision on smaller scales. That significantly improves the final metrics.

---

### Official Review · Reviewer_bPyW · 2023-11-03

**Soundness:** 3 good
**Presentation:** 2 fair
**Contribution:** 3 good
**Rating:** 6
**Confidence:** 3

**Summary:**

This proposes a tensor train decomposition approach to out of distribution detection and approximate nearest neighbor search. The idea is to reshape the Nxd (N vectors in d dims) sample data into (N=N1*N2*...*Nk)xd tensors and compute a tensor train decomposition with lower rank 3-dimensional tensors. That is the data matrix is approximated as G_1 G_2  G_{k-1} where G_i \in R^{r_i x d x i} .  One could fix a prefix of this tensor train and collapse the suffix to find "median" representations of the data while smudging out outliers. A hierarchy of prefixes then would lead to a hierarchy of "median" representations. For ANNS, these could be a hierarchy of cluster centers that the data can be bucketed into (search is then a limited beam search over the tree). For OOD detection, a point cloud is compared to projection via Wasserstein distance. The paper then tries to optimize for approximate objectives and tune over parameters for the two problems.

On the ANN problem, results are compared with a strong clustering based technique called GNO-IMI. A two level hierarchy is chosen and the results closely track those of GNO-IMI.

**Strengths:**

1. This paper demonstrates that tensor train techniques are useful for OOD detection and ANNS.
2. Experiments comparing to GNO-IMI show the technique yields meaningful results.

**Weaknesses:**

1. The paper could be written better. The first few pages are difficult to parse. Many typos too.
2. On ANNS, the paper could have delved deeper. What happens with longer tensor trains -- is it better than GNO-IMI and other techniques or dose this technique saturate? Are there datasets or settings where, notwithstanding its higher compute costs, TT methods are more accurate? While I like the idea of using this new technique, the paper misses an opportunity to draw more conclusions.

**Questions:**

In the ANN section
1. Eq 19: why would the probability not depend on the distance to centroid? This uniform estimate does not look right.
2. Why not compare methods on the number of points in the index the query is compared to (empirically)
3. Can TT methods outperform clustering methods?
4. Can you compare decomposition/index construction times?

typos
abstract: reveals ->reveal
section 2: wide popular->widely popular

---

> ### Author Response · Authors · 2023-11-23
>
> Thank you for your insightful comments and constructive feedback. I truly believe that your suggestions will contribute significantly to making this work better!
>
>
> 1. Equation 19 estimates the number of points to process during the search without prior knowledge about the distribution of the query vector $q$. Thus, if we are given some query vector and the closest centroid is the $i$-th one with probability $p_i$, then we will process size($i$-th centroid) points with probability $p_i$. Because of that, the estimation is $\sum_i p_i N_i$.
> 2. TT can outperform clustering methods. For example, (G)NO-IMI outperforms KMeans for ANN search as index structure, and we showed that the TT point cloud is able to build a better index structure than (G)NO-IMI. But deeper research on the possibility of using the TT point cloud as a clustering method for other use cases is an important future work direction.
> 3. For the sizes of training datasets and TT point clouds used in the paper, training times are quite small and vary by about 10 minutes for the MVTec Ad subsets and 30 minutes for the 10M subset of the Deep1B considered in the paper. We will add comparisons to the main text. Thank you very much for the suggestion!

---

### Official Review · Reviewer_ZRyV · 2023-11-04

**Soundness:** 3 good
**Presentation:** 1 poor
**Contribution:** 2 fair
**Rating:** 3
**Confidence:** 3

**Summary:**

This paper addresses the problem of compressing a database of high dimensional feature vectors (also called point clouds, in the paper).
The compression is performed using a tensor train (TT) low-rank tensor decomposition framework. The authors have motivated their work using primary two applications – (a) out of distribution detection problems (informally also called anomaly detection) and (b) approximate nearest neighbor search. If I understood correctly, then the authors are primarily interested in dealing with large datasets that do not fit in memory and therefore they are interested in aggressively compressing the data such that computation can be directly performed on the compressed TT representation.

In order to compute the tensor train decomposition, the authors have proposed using loss based on comparing distances between distributions (in terms of the Wasserstein distance). They claim that the hierarchical structure in the TT representation enables efficient approximate nearest neighbor search. The OOD detection task is evaluated on the MVTech anomaly detection benchmark where the method is compared with a coreset-based method. For the approximate nearest neighbor detection task, the proposed method is compared with an efficient ANN method proposed by Babenko and Lempitsky 2016.

**Strengths:**

The paper shows that TT decomposition of high dimensional data can be constructed in a way that helps achieve a high rate of compression. The authors then demonstrate that the compressed representation can be useful for two applications -- (1) out of distribution sample detection or anomaly detection, and (2) approximate nearest neighbor search.

**Weaknesses:**

I found the technical methodology presented in the paper difficult to follow and part of the difficulty was that different sections of the paper felt disconnected to each other.
Furthermore, I could not tell what the main contribution is. The tensor train representation itself is well known and has been applied to many different applications involving high dimensional data. There exist several ways to construct such a representation. Section 2 and 3 appears to be a review of relevant known concepts, but the authors did not state clearly that they were reviewing these concepts. For example, Section 2 talks about the tensor train representation of high dimensional data without actually touching upon the crucial aspect of compression. Section 3 finally motivates that different row orders leads to different compression ratio but again I find the discussion a bit vague because it seems like a review of known ideas and insights. The introduction of the sliced Wasserstein loss in Section 3 is a bit abrupt in my opinion. It is unclear why this loss is being defined and why it should be minimized to find the TT parameters. If the main contribution was the introduction of this loss function, then the paper needs to be rewritten to justify why this loss should be used as opposed to the loss function that is conventionally used in existing work.

When considering this as an applications paper, I have some serious concerns as well. Both OOD and ANN are well studied techniques and the baselines against which the proposed TT based approach is compared are not state-of-the-art in the two respective areas. Therefore, I am not convinced about the benefits of using the proposed TT decomposition-based approach to solve either of the two tasks. The experimental results appear to be more of a proof of concept that a TT decomposition-based approach could work for these two tasks.

A few sentences from page 6 (pasted below), shows that the complete Deep1B dataset is not actually used in the ANN experimental evaluation, only a tiny fraction of it is actually used! However, when I checked the paper (Babenko and Lempitsky 2016), I see that the whole Deep1B dataset is used in most of their experiments. Thus, from what I can tell, the ANN results and comparison with G(NO)-IMI presented in this paper, is not a fair comparison, and it raises a serious concern about the practicality of the proposed methods. There are numerous, extremely well-studied, scalable hashing methods for high dimensional data (locality sensitive hashing, learned hashing, etc.) and a more convincing case needs to be made for why this method should be used for approximate nearest neighbor search, instead of such methods.

“We conduct our experiments on the Deep1B Babenko & Lempitsky (2016) – dataset, that contains one billion 96-dimensional vectors, that was produced as an outputs from last fully-connected layer of ImageNet-pretrained GoogLeNet Szegedy et al. (2015) model, then compressed by PCA to 96 dimensions and l2-normalized. In our experiments we use only subset of size 10M vectors.

We build proof-of-concept ANN system based on TT point cloud, … “

**Questions:**

1. I request the authors to clarify the main contribution in the paper and which aspects of the TT decomposition they have used is novel.
2. In the ANN task, what was the reason for not using the complete Deep1B dataset to solve approximate nearest neighbor queries?

---

> ### Author Response · Authors · 2023-11-23
>
> 1. First of all, thank you for insightful comments and constructive feedback. I truly believe that your suggestions will contribute significantly to making this work better.
>
>     The main difference between classical tensor compressions based on Tensor-Train representation and our approach is exactly the independence of the enumeration of samples in the dataset. The standard way of TT application requires constructing the actual compressible tensor, thus fixing a map from the multiindex to the values of the tensor. If a compressible tensor is built explicitly, then this map is an actual multidimensional tensor. If a tensor is built implicitly, it can be determined by some function $(i_1, …, i_k) \rightarrow f(i_1, …, i_k)$.
>
>     This fixed map (fixed order of the vectors in the point cloud) will give poor results for our method. And thus we proposed a probabilistic interpretation of the TT point cloud optimization and then we proposed to use Wasserstein Distance. And this approach allows training TT parameters without predefined ordering of vectors in the database. One can think about it as an automatic implicit adjustment of the points enumeration right during the training procedure.
>
>     The ability of training TT to compress point clouds, without predefined enumeration of points in the point cloud, is our main contribution.
>
> 2. Indeed, we did not use the full Deep1B dataset to solve approximate nearest neighbor queries. The reason for this was the lack of computational possibilities to process the full Deep1B dataset. Instead we performed our tests on the 10M subset of this dataset.
> The comparison with (G)NO-IMI method was performed on the same 10M subset: we retrained this method by ourselves, so we did not use values from the original paper, so the comparison was fair. Although we absolutely agree that for the more  comprehensive comparison it is required to perform tests on the full Deep1B dataset.

---

### Meta-Review · Area_Chair_Ymku · 2023-12-14

**Metareview:**

This paper proposes to utilize tensor-train low-rank tensor decomposition to model point clouds and enable fast approximate nearest-neighbor searches. The paper utilizes Sliced Wasserstein loss to train tensor-train decomposition, and reveals an inherent hierarchical structure for efficient approximate nearest-neighbor searches. The paper further demonstrates the applications on OOD sample detection and approximate nearest neighbor search.

Strengths: The paper demonstrates a tensor-train low-rank tensor decomposition for point clouds, and show its effective applications.

Weaknesses: The paper is hard to follow as commented by all reviewers and the comparison is weak as commented by reviewers.

**Justification For Why Not Higher Score:**

The presentation of this paper is poor and the experimental comparison is weak.

**Justification For Why Not Lower Score:**

N/A

---

### Decision · Program_Chairs · 2024-01-16

Reject